



**X-ray Computed Tomography Investigation of Structures in Opalinus Clay from**
**Large Scale to Small Scale after Mechanical Testing**
Annette Kaufhold[1,2], Gerhard Zacher[3], Matthias Halisch[4], Stephan Kaufhold[1]
(1) Federal Institute for Geosciences and Natural Resources (BGR), Stilleweg 2, D-30655 Hannover,
Germany
(2) Federal Office for Radiation Protection (BFS) , Willy-Brandt-Straße 5, D-38226 Salzgitter, Germany
(3) GE Sensing & Inspection Technologies GmbH, Niels-Bohr-Straße 7, D-31515 Wunstorf, Germany
(4) Leibniz Institute for Applied Geophysics (LIAG), Stilleweg 2, D-30655 Hannover, Germany
**ABSTRACT**
In the past years X-ray Computed Tomography (CT) has became more and more common
for  geoscientific applications and is used from the μm-scale (e.g. for investigations of micro-
fossils or pore scale structures) up to the dm-scale (full drill cores or soil columns). In this
paper we present results from CT imaging and mineralogical investigations of an Opalinus
Clay core on different scales and different regions of interest, emphasizing especially upon
the 3D evaluation and distribution of cracks and their impact upon mechanical testing of such
material. Enhanced knowledge of the testing behavior of the Opalinus Clay is of great
interest, especially since this material is considered for a long term radioactive waste
disposal and storage facility in Switzerland. Hence, results are compared regarding the
mineral (i.e. phase) contrast resolution, the spatial resolution, and the overall scanning
speed.
With this extensive interdisciplinary top-down approach it has been possible to characterize
the general fracture propagation in comparison to mineralogical and textural features of the
Opalinus Clay. Additionally, and as far as we know, a so called mylonitic zone, located at the
intersect of two main fractures, has been observed for the first time for an experimentally
deformed Opalinus sample. The multi-scale results are in good accordance to data from
naturally deformed Opalinus Clay samples, which enables to perform systematical research
under controlled laboratory conditions. Accompanying 3D imaging greatly enhances the
capability of data interpretation and assessment of such a material.

**Key words:** Claystone, μ-CT, Opalinus Clay, Mechanical Testing, HLRW Research
**Corresponding Author:** Matthias Halisch, matthias.halisch@liag-hannover.de



## INTRODUCTION


In the past years X-ray Computed Tomography (CT) has became more and more common
for geoscientific applications and is used from the µm-scale (e.g. for investigations of micro-
fossils or pore scale structures; e.g. Schmitt et al, 2016, this issue) up to the dm-scale (full
drill cores or soil columns; e.g. Schlüter et al, 2015, this issue). Consequently, benchtop CT
equipment for material and geoscience were developed and are now frequently used
because almost all geoscientific samples show 3D features which would be missed when
analyzing 2D sections only (e.g. by classical microscopy). These features are for example
the abundance of minerals, location of particular particles towards bedding (or texture in
general), pore system, cracks, and veins. The 3D distribution of all these features can be
extracted and used for a variety of numerical modeling purposes (Andrä et al., 2013).
However, due to the resolution of µ-CT devices in range of a few µm, it is particularly suitable
to study sandstones or other rocks with large particles and less suitable for the
characterization of clays. Claystones, per definition, feature grain sizes below the common
CT resolution (in range of 1 – 2 µm) and also grain densities, i.e. absorption characteristics,
which result in very challenging segmentation procedures. Although not all features can be
resolved, µ-CT was extensively used to improve the understanding of clays in sediments (oil
industry), in soil science, and as barrier functions in repository systems for high-level
radioactive waste (HLRW). The oil industry is particularly interested in porosity, permeability,
and fluid flow in general. An overview of CT application in the oil industry and for soils is
provided by Heijs et al. (1995) and Akin & Kovscek (2003). By using a medical CT, Ashi et al.
(1997) analyzed texture and density of marine clays, whereas Yang et al. (2010) used CT
data to support logging operations. For soils, Naveed et al. (2012) used CT to investigate the
importance of macropores for the convective fluid flow. The influence of cations on pores of
soils is discussed by Marchuk et al. (2013).
In HLRW research µ-CT was used to investigate the wetting of clay pellets and for the
assessment of homogeneity after wetting (van Geet et al., 2005), relations of mechanical
properties and microstructure (Bésuelle et al., 2006; You et al., 2010), engineering properties
such as deformation (Nakano et al., 2010), and to visualize anisotropy of deformation and
the excavated damage zone (EDZ; You & Li, 2012). Keller et al. (2013) used a set of
different methods (STEM, FIB, and µ-CT) which allowed the "characterization of the pore
structure in the fine-grained clay matrix at different levels of detail" of the Opalinus Clay. The
Opalinus Clay is particularly interesting because it will be the host rock and hence the main
barrier for the Swiss repository for HLRW. In the Opalinus Clay, two different facies can be
distinguished. The clay rich facies is referred to as "shaly facies" and hence distinguished
from the "sandy facies". To resolve differences of both facies, nanotomography was used





(Keller et al., 2013). Micro computed tomography is not suitable to resolve all microstructural
features of clays (micro- and mesopore-range) but rather useful to characterize the
macropore-scale which is relevant for visualizing the crack distribution, advective fluid flow,
and material heterogeneity, such as micro-bedding.
Especially in the field of geomechanical investigations, it is essential to get information about
the mineral composition and microstructure – before and after mechanical tests. All these
parameters have to be characterized to be able to increase the understanding of deformation
processes. While the porosity and microfabric of tectonically undeformed Opalinus Clay
(OPA) (Houben et al., 2013; Keller et al., 2011; Wenk et al., 2008) and naturally deformed
OPA (Laurich et al., 2014) have been intensively studied, little is known of the microstructure
and deformation mechanisms in experimentally deformed OPA.
In this study we present the investigations of an experimentally deformed OPA. The aim is
the visualization of the shear failure in various scales to get more information about the
deformation process. The deformation process is necessary for the long term safety case
analysis for HLRW repositories. Figure 1 showcases the general workflow and the main idea
for the investigation of the Opalinus Clay with a consequent multiple scale (top-down)
approach.

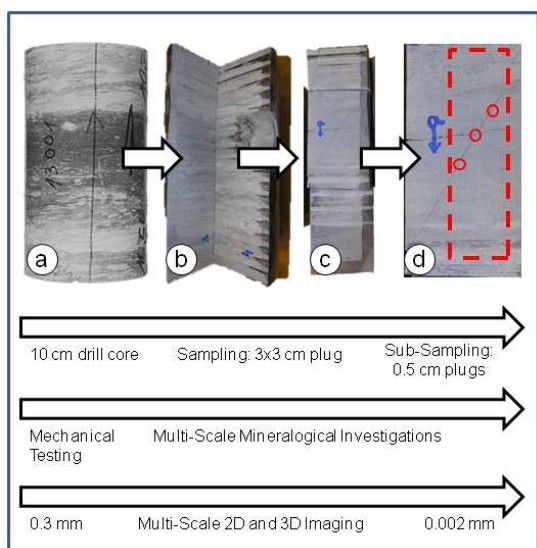


**Figure 1:** Generalized workflow for the multiple scale investigations of the Opalinus Clay: from
mechanical testing and imaging of the 10 cm drill core (a), to sub-sampled plugs (b & c) for
mineralogical and higher resolution imaging, to small scale samples (d) for high resolution and specific
region of interest investigations.



## MATERIAL & METHODS

### Sample Material

The investigated specimen (file 13001, drilling BLT-A6) derives from the Underground Rock Laboratory (URL) Mont Terri, St. Ursanne, Switzerland, and belongs to the sandy facies of the Opalinus Clay (Figure 2). The core sample has a diameter of 100 mm and a length of 180 mm. The drilling is orientated perpendicular to the bedding.

The sample has been sealed by a special vacuum-bag to prevent the material from drying as best as possible, in order to obtain the original saturation condition for the mechanical testing. After the testing it was necessary to stabilize the sample with resin, since shear-failure was fully developed. The specimen was then stepwise sub-sampled for X-ray CT, mineralogical and geochemical investigations on different scales (from dm to mm of sample size).

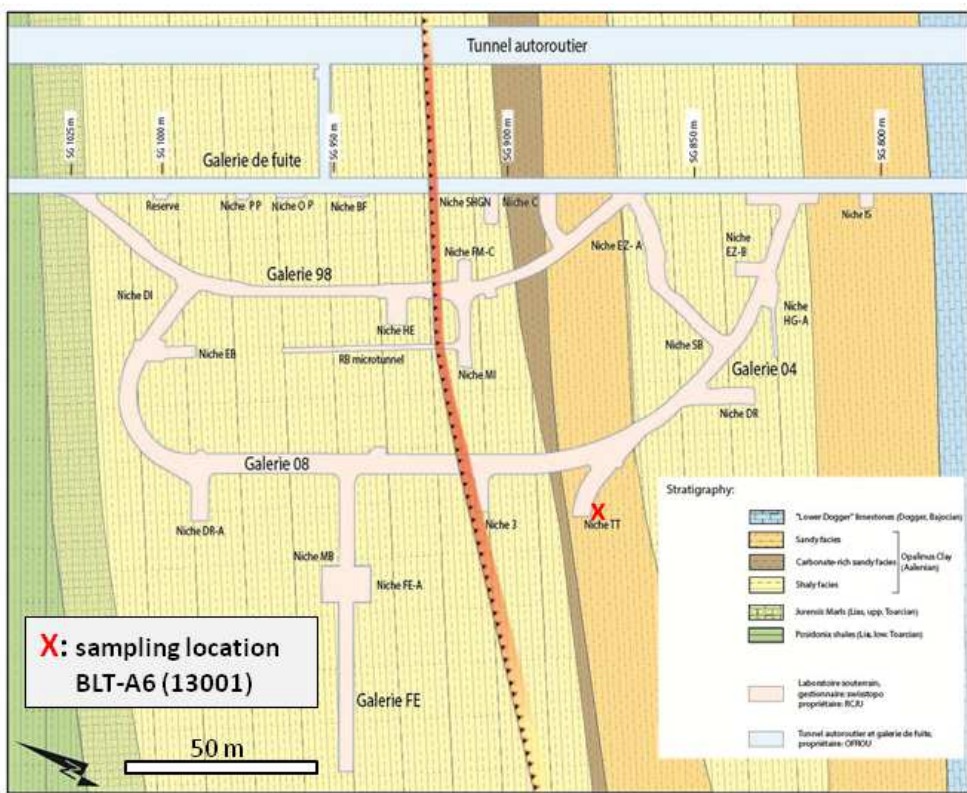

**Figure 2:** Schematic overview of the Mont Terri Underground Rock Laboratory, showing the sampling location of the Opalinus Clay used for this study (modified after XXXX, YYYY).





**Mechanical Testing**

The claystone was tested by triaxial strength testing until a failure was developed. The test was executed in deformation controlled mode with a deformation rate of $d\varepsilon/dt = 10^{-5}$ 1/s and carried out under undrained condition (Gräsle & Plischke, 2010). After the mechanical testing the core was embedded in a resin to stabilize the specimen.

**Mineralogical and Geochemical Investigations**

XRD pattern were recorded using a PANalytical X'Pert PRO MPD Θ-Θ diffractometer (Cu-Kα radiation generated at 40 kV and 30 mA), equipped with a variable divergence slit (20 mm irradiated length), primary and secondary soller, Scientific X´Celerator detector (active length 0.59°), and a sample changer (sample diameter 28 mm). The samples were investigated from 2° to 85° 2Θ with a step size of 0.0167° 2Θ and a measuring time of 10 sec per step. For specimen preparation the top loading technique was used.

For XRF analysis of powdered samples, a PANalytical Axios spectrometer was used (ALMELO, The Netherlands). Samples were prepared by mixing with a flux material (lithium metaborate Spectroflux, Flux No. 100A, Alfa Aesar) and melting into glass beads. The beads were analyzed by wavelength-dispersive XRF. To determine loss on ignition (LOI), 1000 mg of sample material were heated to 1030 ºC for 10 min.

The organic carbon (OC) content was measured with a LECO CS-444-Analysator after dissolution of the carbonates. Carbonates had been removed by treating the samples several times at 80 °C with HCl until no further gas evolution could be observed. Samples of 170-180 mg of the dried material were used to measure the total carbon (TC) content. TIC was calculated by the difference of TC-TOC. The samples were heated in the device to 1800 - 2000 °C in an oxygen atmosphere and the $CO_2$ was detected by an infrared detector. The device was built by LECO (3000 Lake Avenue, St. Joseph, Michigan 49085, U.S.A).

The CEC was measured using the Cu-Triethylenetetramine method (Meier & Kahr, 1999).

 Both the smoothed drillcore section (21 mm x 18 mm) and the three polished core heads (Ø 5 mm) were analyzed for element distribution patterns by an energy-dispersive X-Ray fluorescence spectrometer, the EDXRF microscope M4-Tornado from Bruker-nano. The instrument is equipped with a Rh-tube generating a polychromatic beam, focused by a poly-capillary lens to a spot of a diameter of 17 µm and two Xflash Silicon Drift Detectors (SDD). Take off angle for the tube in moving direction and the detectors is 51° incident and takeoff angle respectively, and the arrangement of the detectors to the tube is in 90° and 270°, respectively. Measuring time was 2 ms at 50 kV, 600 µA and no filters were applied. The



stepsize for the overview was 25 µm and for the core heads 5 µm. False colour evaluation
was performed by using the M4-tornado software esprit.
The polished three drillcore heads were investigated with an Environmental Scanning
Electron Microscope (ESEM, type FEI Quanta 600 FEG) coupled with an energy dispersive
X-ray (EDX) detector (two 30 m² Xflash Silicon Drift Detectors (SDD), Bruker-nano).
Measurement conditions were 25kV, approximately 200 µA, 4 µm spot size, 19 times
magnification at 11.4 mm working distance, and 2 minutes acquisition time.
**X-Ray Computed Tomography**
The Opalinus Clay sample was first scanned with the speed|scan CT 64 located at the GE
facility in Ahrensburg (Germany). Based upon a medical CT system, the CT 64 consists of a
dust protected radiation protection cabinet with an integrated, rotating ring-shaped scanning
device (gantry) and sample transport system for moving components through the scan ring.
The system may accommodate samples of up to 900 mm in length and 500 mm in diameter.
The CT datasets are automatically generated in the so-called helix scan mode with a high
performance rotating anode X-ray tube and a 64 channel multi-line detector rotating around
the sample. Its unique technique allows an overall cycle time of typically 1 minute per
inspected sample (Ambos et al., 2014). Limitation of this type of equipment is the spatial
resolution with typical 0.5 to 1 mm. Nevertheless, due to the very high power of the X-ray
tube (72 kW), different mineral phases can be very distinctively observed. The scan was
recorded with 140 KV and 140 mA within 13 seconds at a spatial resolution of 312 µm.
Second, a CT scan of the same sample was recorded with the v|tome|x L300 system at the
GE facility in Wunstorf (Germany). For technical applications one main goal is the detection
of failures at smallest dimension possible. This approach led to the development of tubes
with small focal spot to enable sharp images at high magnifications. One side effect of this is
that the tube power is hereby limited. The only way to get enough information on the detector
is to increase the scan time, typically from 30 min to 2 hours. In practice this delivers a
resolution of approximately 60 µm for a 10 cm core diameter, which is a factor of 10 better
compared to "medical" CT scanners or such as the speed scan CT. Accordingly, the
differentiation of mineral phases is significantly worse than for the high power system as
described before. The scan parameters were 270 KV and 0.3 mA and the scan duration was
145 min. With this system a spatial resolution of 57.5 µm has been achieved.
For smaller cores (1 to 10 cm) this type of scanning device is still suitable, but when we get
down with sample size to the mm range and thus want to achieve a resolution of a few
microns there is a need to use so called nanofocus tubes with a focal spot size below 1 µm.
For the hereby described studies on 3 mm plugs a nanotom m system (GE Measurement &



Control, phoenix|x-ray) was used. For the 3 cm x 3 cm plug and for the smallest samples
which feature a diameter of 3 mm to 4 mm, a spatial resolution of 2.8 µm has been achieved.
**RESULTS**
**Mineralogical and Geochemical Composition**
The bulk sample is dominated by quartz and carbonates which is typical for the sandy facies
of the Opalinus Clay (Kaufhold et al., 2013; Siegesmund et al., 2013). Amongst the
carbonates calcite was most abundant. In addition, kutnohorite was found which can be
confounded with dolomite because of similar XRD reflections. The existence of traces of
dolomite in addition to calcite and kutnohorite cannot be ruled out. Siderite is present as
trace mineral. Muscovite and illite could not be distinguished because of similar XRD
reflections. Therefore, the presence of muscovite in addition to illite/smectite is possible. The
CEC accounts for 7 meq/100g pointing towards the presence of less than 10 mass-%
smectitic layers which are predominately in illite/smectite mixed layer minerals. Minor
amounts of kaolinite, feldspar, and pyrite were also found.  Using LECO elemental analysis
0.6 mass-% of organic material was found. Assuming an average C-content of carbonate
minerals of about 12 mass-% results in slightly more than 40 mass-% carbonates and 0.9
mass-% of sulfur corresponds to almost 2 mass-% pyrite. This composition is in accordance
with Kaufhold et al. (2013) and Siegesmund et al. (2013).
The aim of the present study was to investigate crack formation which could be related to
microstructural features or mineralogical heterogeneities (as fine bedding, fossil shells, etc.).
Therefore, the heterogeneity was investigated by µ-XRF and SEM. First the crossing of two
cracks was investigated with respect to the mineral indicator elements Si, Ca, Fe, and K. Si
represents quartz, Ca can be mostly found in carbonates, Fe dominates in pyrite and/or Fe-
oxohydroxides, and K indicates clay rich layers because it can be mostly found in
illite/smectite mixed layer minerals being the main clay mineral of the Opalinus Clay. Results
are depicted in Figure 3 . The XRF scanner results reveal the heterogeneities of the sample
in the relevant scale with a resolution of a few µm. The bedding, horizontal in the image, is
reflected by a few mm thick clay layers (green) with more carbonatic layers in between. A
centimeter scaled region was found at the lower left section of the image which could be a
fossil, e.g. a shell fragment. However, this microstructure feature could not be related to the
cracks.





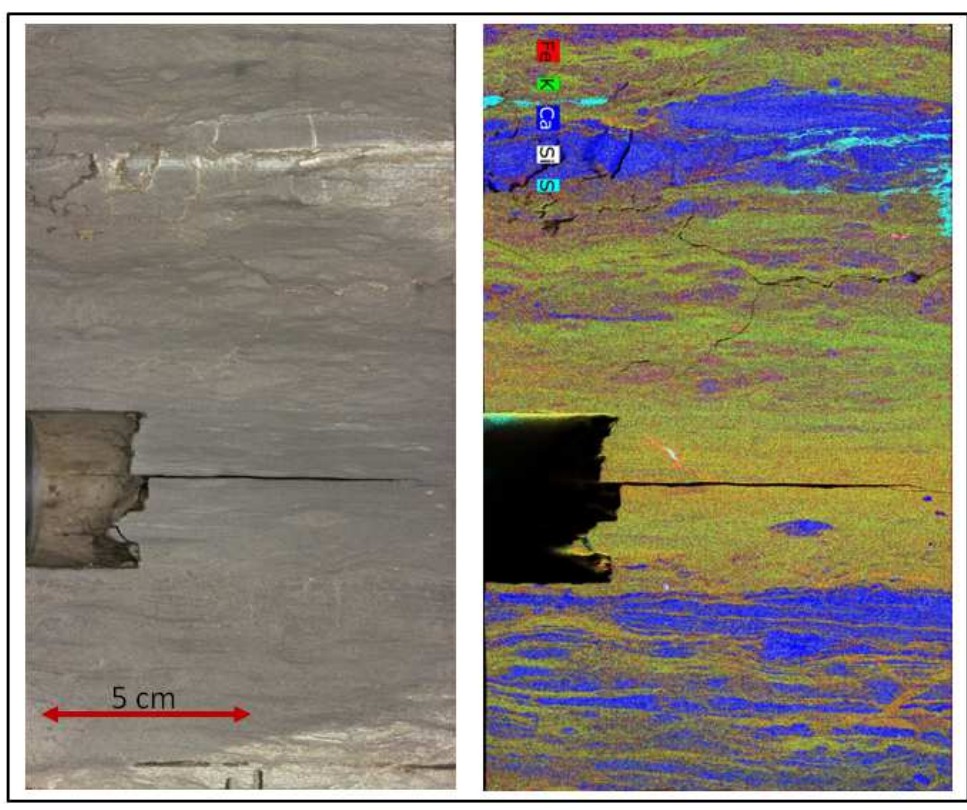


**Figure 3:** Results of the 2D mineralogical mapping on the large core sample (10 cm).
Therefore, magnification was increased (Figure 4). In these small sections of about 5 mm,
bedding features could not be detected anymore. Instead a few 50-100 µm thick bands of
either carbonates (blue) or clays (green) could be observed with a significant angle
compared to bedding. Assuming that these small lineaments were no XRF artefacts, it can
be supposed that a crack started to form there. The location from were tension relief
observable as crack formation started is assumed to be outside the investigated area.
Therefore it can only be assumed that the small lineaments observed both with the XRF
scanner as well as with the SEM could be the starting point for crack formation.





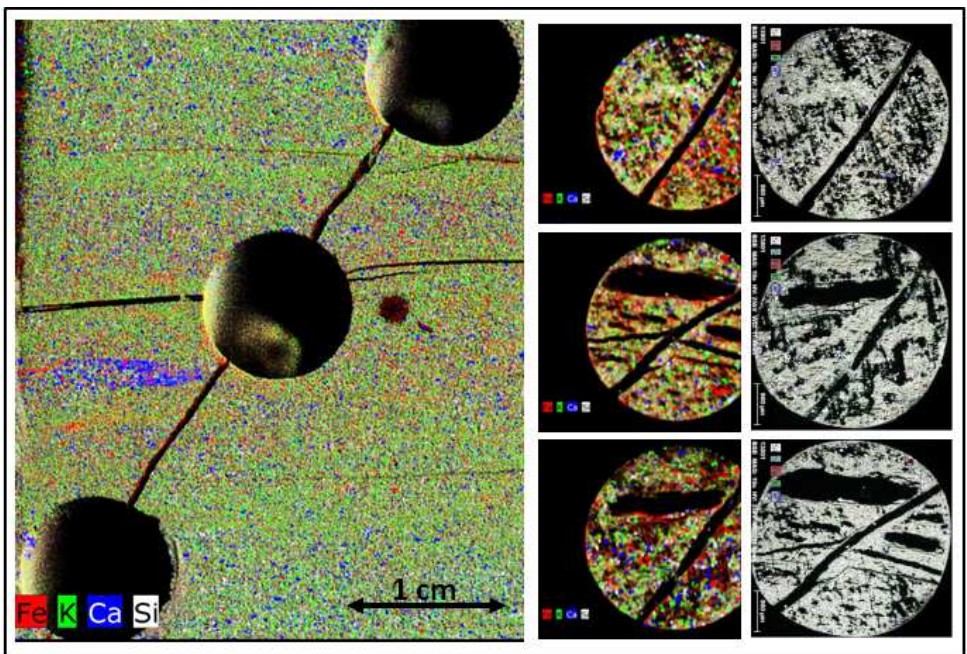


**Figure 4:** Results of the 2D mineralogical and geochemical mapping on the small samples.
**Large Scale and High Speed X-ray CT**
The CT results of the speed|scan CT 64 show good contrast resolution due to its high tube
power (up to 72 kW). Layering, i.e. changes in the mineralogical composition of the core, can
be easily detected based on slightly changing density (see Figure 5). The clay-rich areas are
characterized by darker grey values (e.g. middle section of Figure 5), carbonatic regions are
indicated by higher, i.e. brighter grey values. Layering features can be qualitatively observed
in about the milimeter scale. Cracks and pores can be spatially resolved down to 0.5 mm.
For this core, two main fractures can be observed: a horizontal crack (fracture A), which is
probably caused by de-hydration (so called disking) of the core, and a shear crack caused by
the laboratory mechanical testing. Interestingly, the shear fracture is located within the clay-
rich area of the OPA sample. Starting point is right at the border between clay-rich and
carbonatic zone (right hand side of Figure 5). Additionally, the fracture ends in a carbonatic
region (left hand side of Figure 5) and seems to fan out within that layer. The 3D data set can
be virtually sliced in any direction to emphasize the specific layering or location of the crack
system.





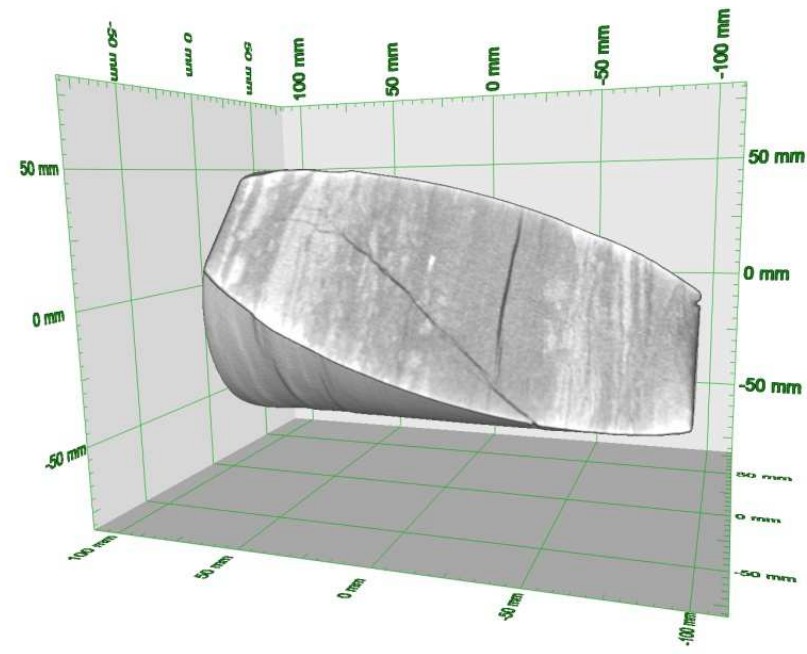


**Figure 5:** Partial 3D view of the speed|scan CT result. Within the virtual core, the layered structure
(due to changes in the mineralogical composition) and two main cracks can be observed.

**Large Scale and High Resolution X-ray CT**

Compared to the faster device explained in the preceding paragraph, the CT results of the
v|tome|x L300 show much better spatial resolution (down to approximately 60 µm for 10 cm
sample width). As the highest power of this system is 0.5 kW, the phase contrast is not as
high but still sufficient to detect larger zones of different densities. On the other hand the
fractures are much better resolved (5 times better resolution) and the delicate network can
be nicely visualized (Figure 6) and studied more in detail. The effective fracture size for
segmentation is in range of the achieved voxel resolution. Segmentation was performed in
the central part of the sample, where the large horizontal crack is intersected by the diagonal
oriented crack system. Additionally, many smaller cracks could be observed, in most cases
also horizontally oriented cracks, which also might be related to disking effects. Interestingly,
a zone of higher fracture density, or at least of higher density due to the lower grey values of
that region, is located near the intersection of the two main fractures (Figure 6, right hand
side). Consequently, this area has been chosen for sub-sampling and 2D and 3D
investigation with higher resolution.




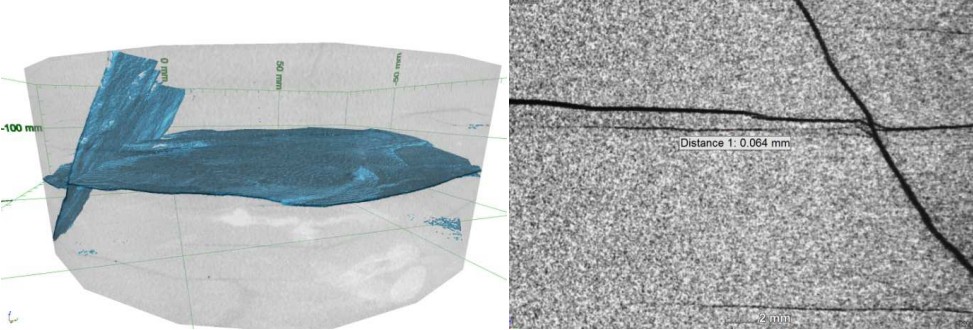


**Figure 6:** The main fractures have been segmented (left hand side) and visualized in a transparent 3D
view. Despite the two major cracks, numerous tiny cracks can be detected. The fracture width can be
measured down to approximately 60 μm.


**Small Scale and High Resolution X-ray CT**


In order to achieve higher image resolution and to obtain good image quality, it is mandatory
to downsize the sample as a smaller voxel size can only be achieved by increasing the
geometrical magnification for the hereby described CT systems. In a first step a 3 cm x 3 cm
sample has been cut out and a CT scan was performed on a nanotom m system with a voxel
resolution of about 18 μm. The results (Figure 7) show significant improvement in diversity of
small details. Individual carbon shells can easily be distinguished and the overall fracture
pattern becomes more and more resolved. Accordingly, smaller shear fractures can be
detected, which are more or less parallel oriented to the main shear crack (Figure 7, left hand
side). The zoomed in view (Figure 7, right hand side) reveals the local presence of small
fractures, which connect the shear cracks with each other.


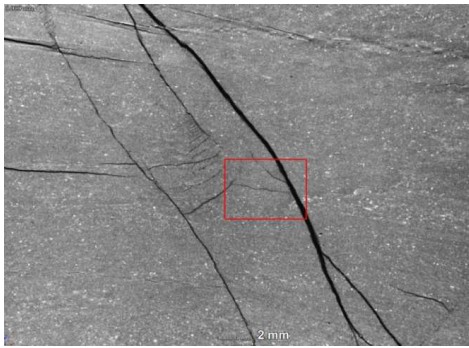
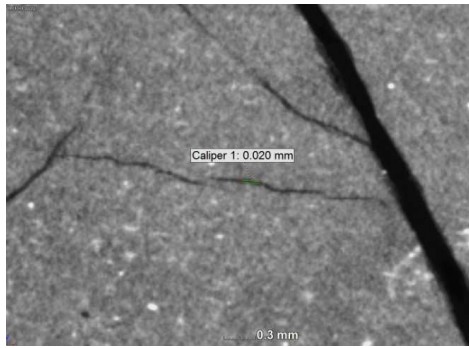


**Figure 7:** Many small scale details of the mineralogical composition are now visible, as well as
numerous small fractures. The red area is zoomed in on the right hand side. Here, a small fracture is
shown, which has an approximate aperture of about 20 μm.






For even smaller scale, micro plugs were drilled with diameter of about 3mm and scanned on
the same system with a voxel resolution of approximately 2 - 3 µm. For this high resolution
single grains can be observed as well as micro cracks and small meso-pores (Figure 8).
Though no specific correlation between fracture occurrence and mineralogy can be
observed, a small zone around that point, where the shear and disking fracture intersect
each other is of very special interest. This area can be characterized as a so called mylonitic
zone, i.e. an area with many small fractures and cracks where particles have been re-
arranged on the fracture surface. As far as literature research reveals, this seems to be the
first reported CT data set of such a zone. For more details, this sample has been used for the
micro-scale mineralogical and microstructure investigations, to achieve more evidence for
this special feature.

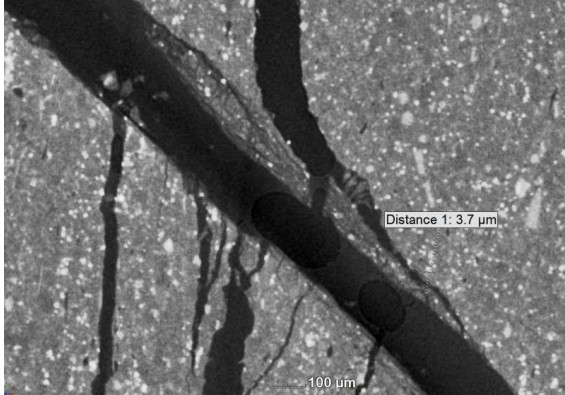
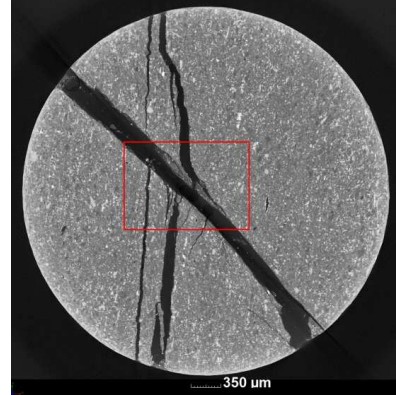

**Figure 8:** For the high resolution data set, many small cracking features can be observed. The
zoomed in area marked by the red box is of special interest, since indications for a mylonitic zone can
be found, where the disking and shear fracture intersect.
**Multi-scale comparison of CT results**
For the investigation of the OPA material, a consequent top-down approach has been used.
Due to the different 3D imaging scales, quantification of sample features (here: the cracks
and fractures) is challenging and may lead to different results. Table 1 highlights the number
of detected cracks and fractures as well as the average aperture of the two main fractures in
relationship to the sample size and to the derived imaging resolution. Whereas the coarse
resolution scans show good results for a first mineralogical and textural sample
characterization, especially details on the fracture development cannot be revealed (Figure

293 9).





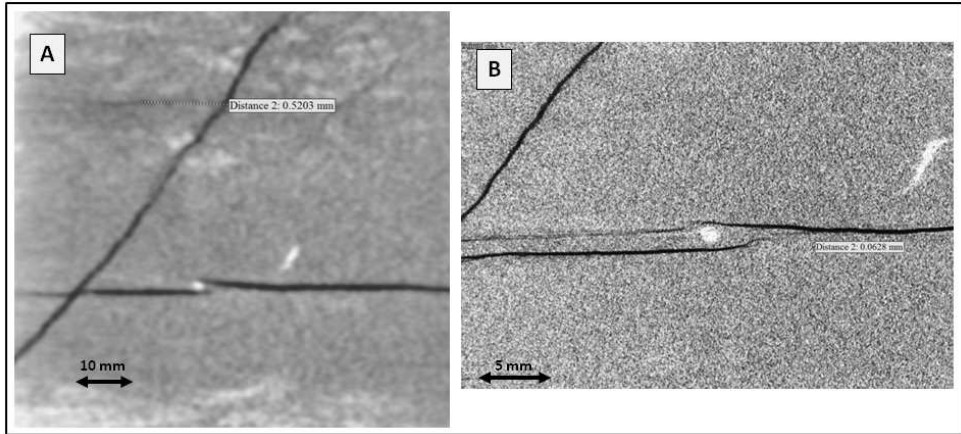


**Figure 9:** Direct comparison of both "coarse" resolution scans with different techniques. Either enhanced phase contrast or spatial resolution can be derived. Hence, both techniques should be used complementary.

Accordingly, sub-sampling the OPA material stepwise greatly increases the information of the fracture network. Hence, the total number of cracks detected increased by a factor of almost 36. If the result is up-scaled to the large core size, this would be a factor as high as 100 to 150. Additionally, the existence of smaller disking as well as of smaller shear fractures has been showcased in the previous sections. As a matter of fact, the evaluation of the two main fractures on different scales leads to very different results. For the large core, fracture apertures are greatly over-estimated (3 to 6 times, related to scanning resolution) due to partial volume effects and – of course – the effective segmentation resolution. Results of the small and micro samples are almost equal for the main crack evaluation. Nevertheless, the number of detected cracks still increases by a factor of about 2. Features such as the observed mylonitic zone can be indicated from the coarse scan data and evaluated in detail by the high resolution image data.

**Table 1:** Comparison of sample size and image resolution related fracture detection and geometrical fracture analysis results for the two main cracks observed.

| core size [mm] | voxel resolution [µm] | # of cracks detected | average crack aperture [µm] | |
| --- | --- | --- | --- | --- |
| | | | fracture A | fracture B |
| 100 | 312,5 | 3 | 990 | 1300 |
| 100 | 57,5 | 15 | 393 | 364 |
| 30 | 17,8 | 47 | 185 | 237 |
| 3 | 2,8 | 107 | 182 | 228 |






**Mircostructural Investigation**


The microstructural investigation was carried out on micro plugs. The plugs were first
scanned with the high resolution X-ray CT. For the scanning electron microscope
investigations (low vacuum) the samples were embedded in resin and the surface was
polished. CT-investigation provided complete 3D information of the samples. One section of
the 3D scan is shown in Figure 10-A. The SEM image, of course, only represents the
polished surface of the plug (Figure 10-B). Dark areas represent cracks filled with air (CT) or
the resin (SEM). Much more dark areas were observed by SEM which resulted from artifacts
caused by sawing and polishing. Ideally an even surface is produced by polishing but the
preparation of even surfaces of claystones is difficult. Depending on sample pretreatment
(drying, wetting etc.) claystones may at least partly disintegrate resulting in a loss of material
upon sawing and polishing. This explains why more dark areas were observed by SEM.
Nevertheless, the main features to be investigated were observed by both techniques.

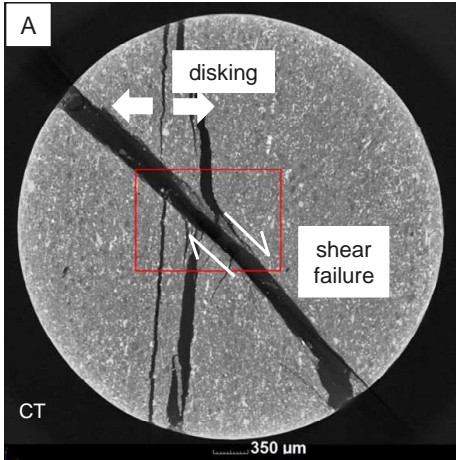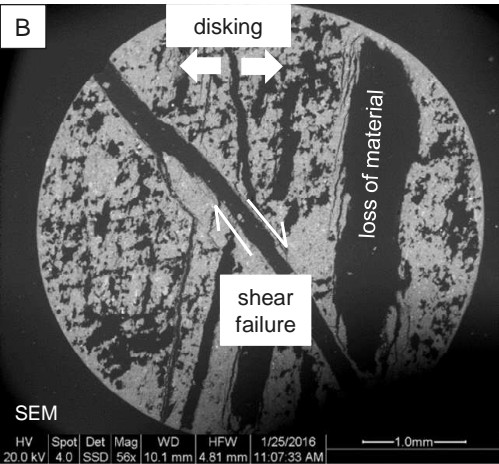

**Figure 10:** Sample overview: A) X-ray CT image of a single layer, B) SEM-image of the polished
surface.
In both figures, shear failure and disking could be observed. Disking is assumed to be a relief
failure in the bedding plane. It was already observed before the mechanical test was
performed. The shear failure is overprinting disking and a material offset is clearly visible in
both images. A closer look at the shear failure reveals more details (Figure 11). In the CT-
image the shear failure and smaller micro-cracks parallel to bedding planes could be
observed. Different minerals can be recognized, but a clear mineral boundary is not visible.
Nevertheless on the top of the shear zone a darker zone is identifiable, which is a result of
particle reduction.



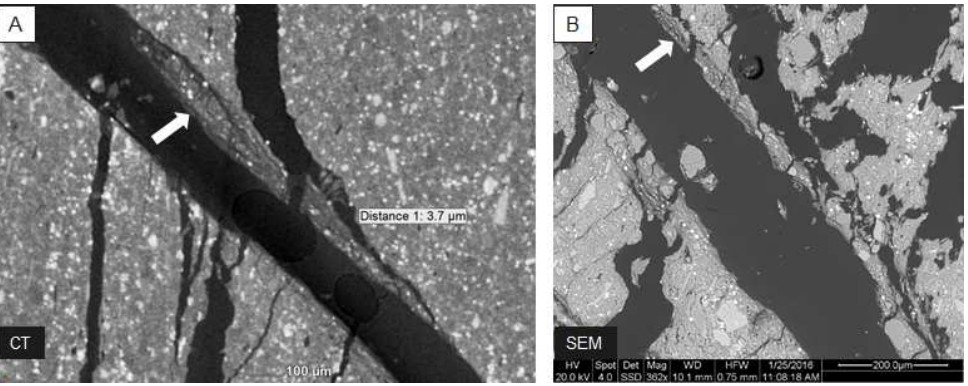


**Figure 11:** Zoomed in view of the shear failure from the X-ray CT image (A) and of the SEM image
(B).
The SEM-image (Figure 11-B) has a higher resolution compared to the CT-image. In
principle, identical microstructure features were found with both methods. In combination with
the 2D SEM inspection, more indications can be observed in order to find out what happened
in the mylonitic zone.

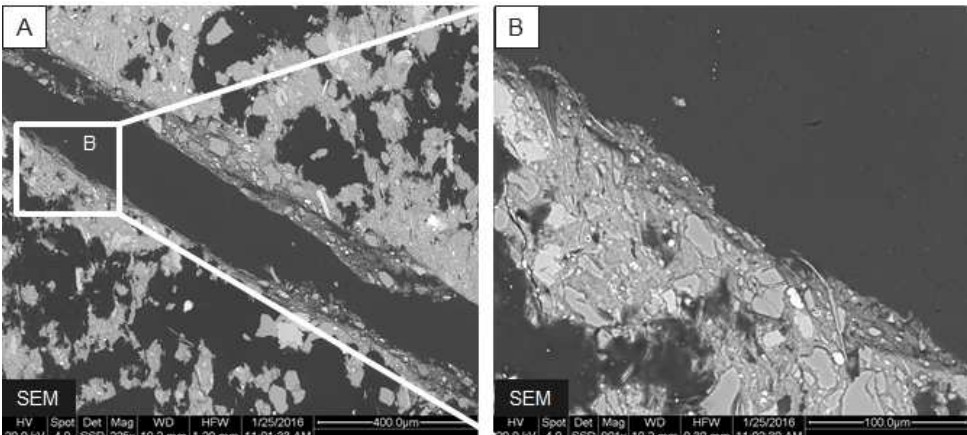


**Figure 12:**   Very high resolution close up of the mylonitic zone located in the shear failure area. It can
be clearly observed, how the particles have been pulled apart from the original matrix material and re-
arranged near the surface.
Close up SEM images (Figure 12) prove that the claystone did not simply break as one
would expect from broken glass. Either before or throughout breaking a rearrangement of the
particles and hence a destruction of the microstructure occurred. Platy particles as micas
rearranged (Figure 12-B) which indicates plastic deformation. As a result a "micro mylonitic"
seam at both sides of the crack was observed. This phenomenon was already observed by
Laurich et al. (2014) for naturally deformed OPA. They explain the occurrence of this
mylonitic zone as a gouge zone. It is not clear whether the mylonitic zone formed just before





breaking or if it formed by the relative movement of both sides of the crack. Nevertheless, it
is a key finding that such a zone exists in artificially deformed OPA, and that this zone has
been observed both, in 2D and 3D data sets.
**CONCLUSIONS**
For the long term safety analysis of repositories for radioactive waste it is necessary to
predict the mechanical behavior of the host rock. The understanding of mechanical
processes in argillaceous rocks is considerably less developed than that of other materials
like salt rocks. Hence the investigations presented above, for microstructure analysis in
various scales regarding mechanical failure, is important to develop our understanding of
mechanical behavior of clay stones.
The OPA material has been intensively studied by a variety of multiple scale and non-
destructive 3D X-ray CT investigations, following a consequent top-down approach to identify
specific regions of interest. According to the mechanical experiment, it has been observed
that the shear failure is located in a clay-rich area. Within the intersecting area of the two
main fractures, a so called mylonitic zone with a particle reduction was observed on the open
shear failure using CT and SEM techniques. But it is not known, until now, when and how
this zone was developed. As far as the authors are aware, this is the first time that
experimental deformation shows such a mylonitic zone.
Therefore it is necessary to investigate further mechanical loaded specimens under different
conditions (water content and strain). These mechanical investigations should be monitored
with non-destructive X-ray CT investigations and in further step accompanied with sub
sampling and small-scale image investigations. Then we have the possibility to get more
information about the petrophysical processes behind the mylonitic zone. All these
investigation can help us to develop our understanding of mechanical behavior which is an
important part in the long term safety analysis of potential hazardous waste disposal places.
**Acknowledgments**
The authors would like to thank Cornelia Müller (LIAG) for the support with the µ-CT imaging
and data set evaluation, Frieder Enzmann (University of Mainz) for filter operation support,
as well as Dieter Rammlmair (BGR) for providing EDXRF and ESEM measuring time and
experience. Furthermore, we would like to thank the reviewers, who helped to improve the
quality of this paper.






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



**Appendix**
**Table 2:** Geochemical and mineralogical composition of the bulk sample.

| mineral composition | | | XRF | | |
|---|---|---|---|---|---|
| quartz | | ++ | $SiO_2$ | [mass-%] | 49,8 |
| calcite | | ++ | $TiO_2$ | [mass-%] | 0,7 |
| Mg-kutnohorite | | +- | $Al_2O_3$ | [mass-%] | 10,7 |
| muscovite/illite (ML) | | +- | $Fe_2O_3$ | [mass-%] | 5,1 |
| kaolinite | | +- | MnO | [mass-%] | 0,1 |
| feldspar | | +- | MgO | [mass-%] | 2,2 |
| pyrite | | +- | CaO | [mass-%] | 12,2 |
| | | | $Na_2O$ | [mass-%] | 0,4 |
| | | | $K_2O$ | [mass-%] | 1,9 |
| | | | $P_2O_5$ | [mass-%] | 0,3 |
| **LECO** | | | $SO_3$ | [mass-%] | 1,2 |
| $C_{total}$ | [mass-%] | 3,6 | LOI | [mass-%] | 15,4 |
| $C_{org}$ | [mass-%] | 0,6 | | | |
| $C_{inorg}$ | [mass-%] | 3,1 | Sum | [mass-%] | 99,9 |
| $S_{total}$ | [mass-%] | 0,9 | | | |
| **CEC** | | | | | |
| CEC | [meq/100g] | 7 | | | |
