# Peer review of "X-ray Computed Tomography Investigation of Structures in Opalinus Clay from Large Scale to Small Scale after Mechanical Testing"

_Solid Earth, 2016_

## Referee Comment (RC1) · W. Graesle (Referee) · 29 Mar 2016

The application of non-destructive X-ray CT techniques for the investigation of geo-materials has become quite common during the last years. Despite the huge number of papers published on this topic, the manuscript appears to be quite unique with respect to the combination of CT techniques on very different scales in a systematic top-down approach (also including additional imaging techniques such as ESEM). The structure of the manuscript is reasonable by simply reproducing the steps of different scale CT-investigations following the top-down order. Overall, this results in an impres-

sive demonstration of the prospects and advantages that CT techniques can provide in the field of rock physics. Applying all investigation techniques to a single core sample which has been subjected to a triaxial strength test adds to the explanatory power of the investigation, because the origin of the analyzed features (shear plane and disking planes) is already well known.

Some minor shortcomings of this manuscript are to be addressed. Most of them will be recovered easily within the process of manuscript revision. In particular, three issues have to be mentioned:

- The results of extensive mineralogical and geochemical investigations are presented. But compared to the excellent combination of results from various CT and imaging techniques into a "synergetic" interpretation yielding a convincing general view, only minor use of the mineralogical and geochemical results has been made in the interpretation.

- The presentation of mechanical processes deserves some improvement, particularly with respect to the discrimination of different failure modes.

- The discussion of features observable in images and 3D data sets is very convincing on the level of qualitative description. In contrast, when attempts are made to derive quantifiable parameters from those data sets, the applied methods are not explained sufficiently and remain somewhat nebulous or even ill-defined. Although those quantifiable parameters only play a minor role in the manuscript, some improvement is recommended.

A detailed listing of various questions and suggestions is added as a supplement (pdf-file)

Please also note the supplement to this comment:
http://www.solid-earth-discuss.net/se-2016-43/se-2016-43-RC1-supplement.pdf

[Figure]

**Supplement:**

Below, questions and suggestions are presented concerning particular passages in the text (referenced by the line numbers in the manuscript):

line 83:

Replacing „The deformation process is necessary for the long term safety case 84 analysis for HLRW repositories"
by "A sufficient understanding of the deformation process is necessary for the long term safety case 84 analysis for HLRW repositories"
would be more precise.

line 98-100:

The importance of sealing for the exclusion of oxygen from the sample to prevent oxidation of pyrite and subsequent formation of gypsum should also be mentioned.

line 109:

The reference for the picture is missing.

line 156-170:

The information concerning the resolution of the employed CT scanners gives a good idea of their capabilities and limitations, but it is not consistent in detail:
The speed|scan CT 64 is characterized by "the spatial resolution with typical 0.5 to 1 mm", whereas the resolution of the performed scan (312 µm) is outside this range.
Consequently, the resolution of the v|tome|x L300 with "approximately 60 µm" is only 5 times better than that of the speed|scan CT 64 – not "a factor of 10 better" as stated in line 166.

line 212:

Replace "were" with "where".

line 208-215:

The given description of Figure 4 (particularly its left part) is only partly comprehensible to me:

"In these small sections of about 5 mm, bedding features could not be detected anymore." –
This is true for the 5 mm sections. But it does not apply to the left part of Figure 4 (showing a pixel size ≤2 times the pixel size in the 5 mm sections). Indeed, on this scale bedding features are no longer dominating the optical impression, but they are still easily detectable. Several layers clearly associated with the orientation of the bedding are shown in an annotated copy of Figure 4. This is worth mentioning because it can provide an indication of principles of crack generation or propagation in this material (see below).

[Figure]

layer of clay + carbonate + quartz mixture

clay layer

layer of clay + carbonate mixture

clay rich layer

layer of clay + carbonate + pyrite
(+ fossile shells oriented parallel
to the bedding planes)

clay rich layer

1 cm

layer of clay + carbonate + quartz mixture

Fe K Ca Si

"Instead a few 50-100 µm thick bands of either carbonates (blue) or clays (green) could be observed with a significant angle compared to bedding." – Locking at the right part of Figure 4, I cannot really identify these bands with a significant inclination with respect to the bedding. If they are there, think on adding some annotation to the figure to assist the observer.

"Assuming that these small lineaments were no XRF artefacts, it can be supposed that a crack started to form there. … Therefore it can only be assumed that the small lineaments observed both with the XRF scanner as well as with the SEM could be the starting point for crack formation." – this appears to be too speculative because there is no evidence available for this assumption. In fact, there is some evidence that crack formation might coincide with material heterogeneities related to bedding features: There are 3 cracks (resp. 3 systems of cracks) in bedding orientation (disking cracks) present in Figure 4; each of them is developed within the layers that exhibit the highest clay content (cf. the annotated copy of Figure 4). Hence, the relative weakness of clay rich layers obviously is relevant for the generation and propagation of cracks in the investigated claystone.

Keep in mind that the 5 mm sections are just 2D images with a rather limited number of pixels. Therefore, some smaller "lineaments" necessarily appear simply by random distribution of mineral particles. Furthermore, there is no evidence that these "lineaments" are structures with a 2-dimensional extension which would be an important requirement for their mechanical relevance. As far as I can see, the only extended heterogeneity visible in the 5 mm sections, which is clearly related to the fractures appears in the first section: The crack separates a clay rich area (left) from a region with high content of pyrite (right). Hence, this boundary between regions of different composition predetermines the propagation of the crack, but there is no evidence that crack formation did start at this feature.

"The location from  where tension relief observable as crack formation started is assumed to be outside the investigated area." – Which cracks does this statement refer to?

- There is one crack orientated diagonally in Figure 4. The formation of this crack most likely started outside the investigated area. But this crack definitely did not originate as a tensile crack (and therefore has nothing to do with a "tension relief"). It is a typical shear failure plane that develops in a triaxial strength test under sufficient confining pressure. The opening of this type of failure plane does not happen until the overall compressive stress state during the strength test is terminated.

- There are 3 cracks (resp. systems of cracks) in bedding orientation present in Figure 4. These cracks are most likely formed by tensile failure. As far as I can see, there is no evidence that the formation of these cracks started outside the investigated area. In contrast, the observation that these cracks split into systems of interconnected subparallel cracks in the vicinity of the shear failure plane (cf. Figure 4) might offer some indication that their formation started at the shear failure plane. Regarding the small scale heterogeneity of the stress field occurring along the shear failure plane during shearing (due to inhomogeneous friction as well as to unevenness of the shear failure plane), an initiation of tensile cracks in this area seems likely.

Thus, whatever crack might be addressed in line 212/213, to me the statement appears to be unconvincing (for different reasons depending on the addressed crack).

line 216:

The last 2 pictures in the right side column have to change places to correspond to the pictures in the central column.

line 227-229:

"Interestingly, the shear fracture is located within the clay rich area of the OPA sample. Starting point is right at the border between clay carbonatic zone (right hand side of Figure 5)." – This statement, in particular the second sentence, cannot be proven by a 2D-picture as it is presented by the virtual slice in Figure 5 (the intersect of the failure plane with the cylinder barrel could extend considerably into the carbonatic zone in parts not visible in Figure 5). But checking "Video 3 - speed scan data set.avi" in the supplement proves that the statement is correct. Thus, adding a reference to the video in this context might be a good idea.

line 253:

"Besides" instead of "Despite" appears more appropriate.

line 261:

Replace "carbon shells" with "carbonate shells".

line 262-264:

"Accordingly, smaller shear fractures can be detected, which are more or less parallel oriented to the main shear crack (Figure 7, left hand side)." – There are at least 3 types of cracks visible in Figure 7:

- The long diagonal cracks which sometimes split up into several subparallel braches. These are the shear fractures constituting the shear failure plane.

- Some cracks following bedding plane features. Most likely, these have been formed by tensile failure.

- A system of small stacked cracks orientated more or less perpendicular to the large shear fractures. They have obviously formed later than the large shear fractures as they are truncated by the shear fractures. Generally, shearing normal to the main shear failure plane is expected to be almost negligible. Therefore, these stacked cracks are very likely generated by tensile failure.

This might be explained as result of a slight bending, that can affect the thin block of material between 2 large parallel shear fractures during the shearing.

line 298-301 and Table 1:

Generally, the conclusion of this statement is comprehensible and plausible. However, the methodic approach for the quantifying statements accompanying the argumentation is not at all trivial. Since the applied method for counting cracks and evaluating their frequency is not explained in the text, the basis of the argumentation remains nebulous and not well-defined. A number of questions arise regarding the determination of the "number of cracks" given in Table 1:

- Are these numbers derived from 2D-slices or from full 3D data?
- There is only one linear dimension ("core size") given in Table 1. Hence it remains unclear, to which area (in case of 2D-slices) resp. volume (in case of full 3D analysis) the given crack numbers refer. Do all crack numbers refer to the same area resp. volume of the sample? This would restrict the analysis to a very small detail of the low-resolution scans. But it is mandatory to preserve comparability of the numbers, because otherwise scaling problems and effects of non-representative subsampling would make meaningful comparison of crack numbers almost impossible.
- Which criteria have been used to establish a well-defined method for counting cracks?

Basically, it has to be questioned whether the "number of cracks" is an appropriate measure to evaluate and compare the amount of information yielded by different CT techniques. First of all, a number of cracks has to be identified in a specified area resp. volume. Thus, a "crack number density" would be a more appropriate information. But there are still serious problems with this approach. Obviously, there is a scaling problem: Because cracks often range beyond the boundaries of an investigated volume, the number of cracks will not grow proportionally when increasing the investigated volume. Thus, the "crack number density" is a scale dependent quantity.

Another, even more severe problem arises from the fact that the determination of a "crack number density" inevitably requires the determination of a "number of cracks". When using the term "number of cracks", well-defined criteria are required for what has to be counted as "1 crack". A consistent definition how to distinguish between "1 crack" and "2 or more associated cracks" is extremely difficult to achieve even in 2D. You can easily illustrate this problem by asking several people to determine the number of cracks visible in the left picture of Figure 8 – you will get a considerable variation between their answers. When switching to a 3D analysis this problem becomes even more severe.

Regarding these shortcomings, it is recommended to employ another measure to characterize the amount of information yielded by different CT techniques. This measure should exhibit considerable advantages compared to the "number of cracks" or the "crack number density". In particular, it has to be determinable in a well-defined way, and it should be virtually unaffected by scaling effects. Whereas counting cracks turned out to be very problematic, the determination of crack area is a largely straightforward procedure offering the required advantages. Therefore, I recommend to use a "crack area density" (i. e. the area of detected cracks per investigated sample volume) instead of the "number of cracks".

line 302-306:

It is evident that voxel resolution will have a considerable impact on determining the dimension of features that are close to or even below the voxel size. Nevertheless, the shown degree of this impact is surprising.

Partial volume effects are not sufficient to explain the amount of overestimation of crack aperture

occurring in the large core scans: Even if voxels containing an almost negligible portion of crack are classified as part of the crack, the overestimation of the aperture $w$ of a straight crack is limited to 2 times the voxel size: $w_{obs} \leq w_{true} + 2\, d_{voxel}$. On average, the impact of partial volume effects should be much smaller. Assuming that the average aperture determined on 3 mm core size represents the true value, the average aperture observed on 100 mm core exceeds this theoretical limit explainable by partial volume effects in any case (see table).

| | | average crack aperture [μm] | | | |
| | | fracture A | | fracture B | |
| core size [mm] | voxel resolution [μm] | observed | upper limit regarding partial volume effects | observed | upper limit regarding partial volume effects |
|---|---|---|---|---|---|
| 100 | 312.5 | 990 | 807 | 1300 | 853 |
| 100 | 57.5 | 393 | 297 | 364 | 343 |
| 3 | 2.8 | 182 | | 228 | |

Therefore, a significant part of the overestimation must be attributable to the impact of the effective segmentation resolution. Since this can vary significantly depending on the settings for numerous data processing parameters, adding another column to Table 1 showing the effective segmentation resolution is advisable.

line 310-312:

Identifying the fractures as "shear crack" and "disking crack" would be more informative than just numbering them "fracture A" and "fracture B".

line 335-336:

"Nevertheless on the top of the shear zone a darker zone is identifiable, which is a result of particle reduction." – Since X-ray attenuation does not depend on particle size (as long as particle size is large compared to the wavelength of the X-ray), but only depends on material density and the mineral composition, the darker zone should not be explained as "a result of particle reduction". It is rather a result of loosening of the material (dilatancy) resulting in a lower density. Nevertheless, it is obvious from the overall impression of Figure 11-B and Figure 12 (and should hence be mentioned in the text) that this dilatant deformation of the material is accompanied by a reduction of particle size.

line 348-349:

"Close up SEM images (Figure 12) prove that the claystone did not simply break as one would expect from broken glass." – This statement should be presented in a less generalized and more precise form.
The term "break" does not differentiate between shear failure and tensile failure, although these represent completely different failure mechanisms. As can be recognized from Figure 11, a mylonitic zone is only formed along shear cracks, whereas tensile cracks do not show any mylonitic features. Therefore, the statement in line 348-349 only applies to the breaking of claystone under deviatoric loading (shear). The statement is wrong with respect to tensile failure in claystone, which looks quite similar to the pattern one would expect from broken glass.
Due to the missing differentiation between different failure regimes, the given comparison between breaking claystone and breaking glass is misleading. One has to be aware that almost any case of breaking glass we know from our everyday experience represents a pure tensile failure mode. Thus, the difference in features observable on failure cracks, which is attributed to a

material difference (claystone vs. glass) in the text, in fact is attributable to different failure modes (shear vs. tension).

line 354-355:

"It is not clear whether the mylonitic zone formed just before breaking or if it formed by the relative movement of both sides of the crack." – Taking into account the amount of local deformation and particle dislocation required to form the mylonitic zone, a formation before breaking would be very difficult to explain.

---

## Referee Comment (RC2) · Anonymous Referee #2 · 5 Apr 2016

Title: X-ray Computed Tomography Investigation of Structures in Opalinus Clay from Large Scale to Small Scale after Mechanical Testing

General comments Thank you very much for this paper. It match's perfect into the scope of the special issue in Solid Earth. This paper should be a welcome addition to the technical literature in the field, and its scientific content is quite good. I am interested in "Mirco-structural Investigation" on line 313 in page 14 because the both result supplemented lost data each other. These results will be able to lead the idea to progress the discussion for micro-macro issue on material deformation/failure. Mean-

while, still some modification should be required. Please check some questions and comments like below.

Specific comments Line 23 in page 2 To-down approach will give wrong direction about this paper's concept. It should be scale down approach.

Line 45-47 in page 2 Probably, the authors' background is rock engineering or geological engineering? This is partially correct but not exactly. This sentence could be sandy soil, sandstone or other rocks. This part is more rather general topic so it may be good to include more general material like soil as well.

Line 62 in page 2 This reference seems to be conference paper. Probably, this reference paper has been upgrade as Toshifumi Mukunoki, Takahiro Nakano, Jun Otani and Jean-Pierre Gourc (2014), Study of cracking process of clay cap barrier in landfill using X-ray CT, Applied Clay Science, Vol. 101, pp. 558-566, DOI 10.1016/j.clay.2014.09.019

Line 94 in page 4 What are these (file 13001, drilling BLT-A6) information?

Line 101 in page 4 How authors did conduct this work? How can you keep the failure condition after testing? It is better to explain more because this approach is important but not so easy, I imagine.

Line 108 in page 5 Do you mean "triaxial compression testing"? it is better to show the picture of testing scene. If you did triaxial compression test, how much the confining pressure? Was the samples saturated? The information of this session is very little. Authors should add more testing condition.

Line 126 in page 5 Is this hydrogen chloride "solution"? Is there any information about concentration? It is necessary to explain the concentration.

Line 166 in page 6 which is a factor of 10 better compared to. . ... What do you mean "10" here?

Line 193 in page 7 "The aim of the present study was to investigate crack formation which could be related to microstructural features or mineralogical heterogeneities (as fine bedding, fossil shells, etc.)." This sentence should be appeared in the beginning of this section to clarify the authors concept.

Line 243 in page 10 "Segmentation" How do you segment the image? It is necessary for authors to discuss threshold value here.

Line 256 in page 11 "achieve" "achieve "replace other verb. How do you achieve higher image resolution? If authors have special idea, please explain the meaning of "achieve higher image resolution.

Line 277 in page 12 Please add the some reference which authors referred.

Line 288 in page 12 this seems to be the first reported CT data set of such a zone This sentence is not clear about author's point. Please explain more.

Line 299-301 in page 13 Hence, the total number of cracks detected increased by a factor of almost 36. If the result is up-scaled to the large core size, this would be a 100 to 150. We are not sure the mean of these values. What are they? Please explain more.

Line 313 in page 14 Mircostructural Investigation Your title is "X-ray Computed Tomography Investigation of Structures in Opalinus Clay from Large Scale to Small Scale after Mechanical Testing". The main tool of this paper is CT so it would be better to summarize this chapter by the observation obtained from CT in the last paragraph. Maybe I am wrong because authors described the aim of this study is the visualization of the shear failure in various scales to get more information about the deformation process on line 81-83 in page 3; however, I am concerned about the structure of last chapter. It may be better if authors summarize the conclusion with more CT data.

Line 356-357 in page 16 What do you mean 3D? You are showing 2D image only here (see Figure 10, 11 and 12)

Line 374 in page 16 This paper did not mention about any water contents

Technical correction about text Line 68 in page 2 facies "is", this could be "are"?

Line 98 in page 4 The "sample has" should be "samples have"?

Line 119 in page 5 X-ray fluorescence(XRF) spectrometer

Line 127 in page 5 What is TIC? This may be TC?

Line 128 in page 5 What is TOC?

Line 129 in page 5 CO2 should be $CO_2$.

Line 131 in page 5 The CEC should be "The Cation Exchange Capacity (CEC)" .

Line 147 in page 6 "X-Ray Computed Tomography" should be "X-ray Computed Tomography (CT)".

Line 159, 169 in page 6 140KV and 270KV should be 140 kV and 270 kV.

Line 180in page 7 Opalinus Clay should be Opalinus (OPA) clay

Line 212 in page 8 some words may be missed here after "from".

Line 214 in page 8 It is better to use other conjunction because you used same conjunction in same paragraph already.

Line 262 in page 11 smaller than what?

Line 286 in page 12 Top-down should be Scale done?

Line 315 in page 14 Put (SEM) after "For the scanning electron microscope"

Technical correction about Figures and Tables Figure 1 in page 3 What is the point of red part in Figure 1? Explanation is very few so some more explanation should be added.

Figure 2 Probably, this is because of PDF conversion? Anyway, the quality of Figure 2

is not great.

Figure 4 Put "(a), (b)" to each image. Do you need image on the right lane?

Figure 5 What is authors' point for Figure 5?

Figure 6 Is Figure 6 a part of Figure 5? This should be explained more. Put a scale in the right Figure.

Figure 7 Scale is not clear in left image. Caliper and scale are not clear on right image.

Figure 8 Where did authors measure in Figure 8. It is not clear about Distance 1:3.7 ïA▪m. The scale in Figure 8 is not clear.

Figure 9 Same comment as Figure 8 in the above. Put red line in left image like Figure7 and 8. What do authors want readers to focus?

Figure 10 Use not red square but red circle in left image to compare the essential part.

Figure 11 Same comment as Figure 9.

Table 1 Delete first and last line in the table 1.

---

## Author Comment (AC1) · 13 Apr 2016

**REVIEWER 1**

Line 83: implemented as mentioned

Line 98-100: has been added

Line 109: *reference was included in the revised version.*

Line 156 – 170: thanks for the advice. We have rephrased this part of the manuscript.

Line 208-205: *good comment – text added to the revised version*

Line 212: changed as mentioned

Line 212 – 215: *the reviewer is right in pointing out that the interpretations of the observations are more difficult than yet explained. A detailed discussion, however, would be highly speculative. Therefore, the interpretation is relativized:*
*the red marked text was added behind* "Assuming that these small lineaments were no XRF artefacts, it can be supposed that a crack started to form there. … Therefore it can only be assumed that the small lineaments observed both with the XRF scanner as well as with the SEM could be the starting point for crack formation.

This interpretation, however, only represents an option. As an alternative crack formation may also coincide with material heterogeneities such as bedding features. Observations presented here do not allow for an unambiguous conclusion because the crack apparently formed outside the investigated area.

In addition, the figure will be modified as recommended!

Line 216: correct, figures have changed places and now fit.

Line 227-229: good advice, thanks! Implemented as mentioned

Line 253: changed as mentioned

Line 261: changed as mentioned

Line 262-264: *Reviewer requested a more detailed explanation of the features which could be resolved. Sentence* "the zoomed in view…" *was replaced by the following text:*

At least three types of cracks could be distinguished based on the improved resolution imaging. Long diagonal cracks which sometimes split up into several subparallel braches were observed. These are the shear fractures constituting the shear failure plane. Secondly, some cracks following bedding plane features were observed. They may have formed by tensile failure. Finally, a system of small stacked cracks orientated more or less perpendicular to the large shear fractures could be observed. They may have formed later than the large shear fractures as they are truncated by the shear fractures. Generally, shearing normal to the main shear failure plane is expected to be almost negligible. Therefore, these stacked cracks probably formed by tensile failure.

Line 298-301 & table 1: good points, thanks. We will add substantially details as also recommended by reviewer 2 about the crack counting workflow and quantification, in order to clarify the points mentioned

Line 302-306: In fact, the large dispersion is caused by three different effects:

- By "resolution" (i.e. partial volume effects)
- By the "effective segmentation resolution" and
- By the fact that we are looking at different sample volumes and hence "miss" larger apertures as we increase the resolution and accordingly decrease the volume of interest (due to the limited field of view of the CT devices)

We will describe this section and the workflow and effects a little more in detail in order to fit your recommendations properly.

Line 310-312: *good comment – it's much clearer to refer to 'shear cracks' and "disking crack'.*

Line 335-336: *the reviewer requested to be more precise in this sentence:* "Nevertheless on the top of the shear zone a darker zone is identifiable, which is a result of particle reduction."

*This sentence was replaced by:*

"Nevertheless on the top of the shear zone a darker zone was observed which may have resulted from a loosening of the material resulting in lower density"

Line 348-349: The reviewer requested to be more precise in the following sentence:

"Close up SEM images (Figure 12) prove that the claystone did not simply break as one would expect from broken glass." The following text was added: "Such an appearance of the cracks points towards shear failure. Cracks induced by tensile failure would miss a mylonitic zone and hence be closer to "broken glass"."

Line 354-355: the reviewer requested to be more precise in the following sentence:

"It is not clear whether the mylonitic zone formed just before breaking or if it formed by the relative movement of both sides of the crack." The following text was added: "However, taking into account the amount of local deformation and particle dislocation required to form the mylonitic zone, a formation before breaking would be very difficult to explain."

**REVIEWER 2**

Line 23: Seems more or less to be the same for us, but we have changed it as recommended.

Line 45-47 in page 2: Probably, the authors' background is rock engineering or geological engineering? This is partially correct but not exactly. This sentence could be sandy soil, sandstone or other rocks. This part is more rather general topic so it may be good to include more general material like soil as well.

*"and soil" was added*

Line 62 in page 2: This reference seems to be conference paper. Probably, this reference paper has been upgrade as Toshifumi Mukunoki, Takahiro Nakano, Jun Otani and Jean-Pierre Gourc (2014), Study of cracking process of clay cap barrier in landfill using X-ray CT, Applied Clay Science, Vol. 101, pp. 558-566, DOI 10.1016/j.clay. 2014.09.019

*Ref was added*

Line 94 in page 4 What are these (file 13001, drilling BLT-A6) information?

*Explanation was added*

Line 101 in page 4 How authors did conduct this work? How can you keep the failure condition after testing? It is better to explain more because this approach is important but not so easy, I imagine.

*This is an important point. In line 100 it is explained that the sample was stabilized by a resin directly after testing.*

Line 108 in page 5 Do you mean "triaxial compression testing"? it is better to show the picture of testing scene. If you did triaxial compression test, how much the confining pressure? Was the samples saturated? The information of this session is very little. Authors should add more testing condition.

*The following information was added: "confining pressure 6 MPa, natural water content was preserved"*

Line 126 in page 5 Is this hydrogen chloride "solution"? Is there any information about concentration? It is necessary to explain the concentration.

*"solution" was added*

Line 166: The factor of 10 refers to the scanning resolution.

Line 193 in page 7 "The aim of the present study was to investigate crack formation which could be related to microstructural features or mineralogical heterogeneities (as fine bedding, fossil shells, etc.)." This sentence should be appeared in the beginning of this section to clarify the authors concept.

*Was moved*

Line 243 "Segmentation"

We have added a brief description about the segmentation procedure.

Line 256: we replaced "achieve" with "reach"

Line 277 in page 12 Please add the some reference which authors referred. / Line 288 in page 12 this seems to be the first reported CT data set of such a zone. This sentence is not clear about author's point. Please explain more.

*We are not aware of any suitable references. Maybe you can provide some?*

Line 299-301 "number of cracks".

Well, we have some doubt that "upscaling" to large cores is that easy. Hence we avoided to extrapolate any data about this. The crack number detection has just been performed in order to outline the importance of resolution and detail detectability as a "function" of investigation scale. We will be a bit more precise about our intention within this section and will rephrase slightly.

Line 313 in page 14: Mircostructural Investigation Your title is "X-ray Computed Tomography Investigation of Structures in Opalinus Clay from Large Scale to Small Scale after Mechanical Testing". The main tool of this paper is CT so it would be better to summarize this chapter by the observation obtained from CT in the last paragraph. Maybe I am wrong because authors described the aim of this study is the visualization of the shear failure in various scales to get more information about the deformation process on line 81-83 in page 3; however, I am concerned about the structure of last chapter. It may be better if authors summarize the conclusion with more CT data.

*About the same was requested by Rev1, more explanations were about visualization of cracks were added to the revised version. In addition, the title of the chapter was modified according to R2's statement.*

Line 356-357: Well, we showcase 2D images, nevertheless, the quantification of the structures has been performed on the 3D data sets.

Line 374 in page 16: This paper did not mention about any water contents

*In the revised version it is stressed that the natural water content of the Opalinus clay was preserved. A determination of the water content was not possible because the water content of an undisturbed sample had to be preserved and after the tests the sample had to be sealed by a resin.*

General comments Reviewer's pages C4-C5:

Points have all been addressed